# Porphyrins as Chelating Agents for Molecular Imaging in Nuclear Medicine

**DOI:** 10.3390/molecules27103311

**Published:** 2022-05-21

**Authors:** Krystyna Pyrzynska, Krzysztof Kilian, Mateusz Pęgier

**Affiliations:** 1Faculty of Chemistry, University of Warsaw, Pasteura 1, 02-093 Warsaw, Poland; 2Heavy Ion Laboratory, University of Warsaw, Pasteura 5A, 02-093 Warsaw, Poland; kilian@slcj.uw.edu.pl (K.K.); pegier@slcj.uw.edu.pl (M.P.)

**Keywords:** porphyrins, metallic radionuclides, single-photon emission computed tomography, positron emission tomography

## Abstract

Porphyrin ligands, showing a significant affinity for cancer cells, also have the ability to chelate metallic radioisotopes to form potential diagnostic radiopharmaceuticals. They can be applied in single-photon emission computed tomography (SPECT) and positron emission tomography (PET) to evaluate metabolic changes in the human body for tumor diagnostics. The aim of this paper is to present a short overview of the main metallic radionuclides complexed by porphyrin ligands and used in these techniques. These chelation reactions are discussed in terms of the complexation conditions and kinetics and the complex stability.

## 1. Introduction

In contemporary medicine, an important role is played by non-invasive diagnostics, allowing early detection of disturbances or pathological changes due to disease. Recently, in this field, the role of imaging methods and non-invasive tracking of physiological processes on the molecular level has significantly increased. One of the observed phenomena is the increasing importance of nuclear medicine. These methods, in contrast to popular anatomical imaging methods, use radioisotopes administered to the patient to track the distribution of labeled substances. This means that the patient is not exposed to a sampling factor from an external source (i.e., in X-ray examination), but imaging is achieved after administering a radioactive substance that generates radiation inside the patient’s body and then is recorded by a system of external detectors. This approach allows the biodistribution of the administered substance to be tracked in the examined system or organ and offers qualitative and quantitative information about the concentrations, dynamics, and accumulation in target organs [1]. 

SPECT (single-photon emission computed tomography) uses radioactive gamma isotopes for functional and molecular imaging [2]. The effective range of the radiation energies used is 50–300 keV and the gamma quanta are recorded in planar detectors with collimators, defining the region of the target signal acquisition by a single detector. In SPECT, thanks to the use of several rotating heads with detectors, it is possible to obtain 2- or 3-D images, both static (e.g., images of pathological bone changes (scintigraphy)) and dynamic (imaging of blood flow in the brain (perfusion)). In healthy people, the distribution of the radiopharmaceutical follows the physiological pattern while in the case of dysfunction, regions with increased or decreased contents are identified. 

Positron emission tomography (PET) is a highly sensitive nuclear medicine imaging technique capable of observing metabolic changes in the human body in real time. It utilizes substances labeled with short-lived radionuclides that are injected intravenously into a patient and are distributed within different tissues according to the carrier molecule. Radionuclides used in PET undergo beta plus (β^+^) decay, which leads to emission of positron (e^+^). Positron, after losing kinetic energy, interacts with an electron in an annihilation reaction, resulting in the emission of two gamma photons at an angle of 180 degrees with a characteristic energy of 511 keV. After detection of these coincidences, the distribution of the radiotracer is calculated by determining the point of annihilation. The major application of PET is clinical oncology, but it is also applied in the diagnostics of cardiac diseases and neurological disorders [3,4]. The relatively short half-lives of radionuclides allow the tracers to reach equilibrium in the body, which especially includes accumulation in the molecular target and washout from other tissues, but without exposing the subjects to prolonged periods of radiation.

Construction of a radiopharmaceutical usually involves the conjugation of the radioactive isotope to the targeting molecule using a bifunctional ligand [4,5]. The bifunctional ligands are able to coordinate the radionuclide and simultaneously are covalently attached to the targeting molecule (small proteins, peptides, fragments of monoclonal antibody) either directly or through various linkers. The target molecules are able to transport complexed radionuclide to the diseased tissue containing the appropriate target receptor or include it in a specific biochemical pathway. The complexes used in nuclear medicine have to exhibit a high thermodynamic stability as a strong interaction between the metal and the ligand is necessary to ensure the complete complexation of the radionuclide. Moreover, a radioisotope–ligand complex should exhibit high kinetic inertness to prevent the dissociation of the complex and, thus, the release of the radionuclide in the biological medium.

A wide variety of acyclic and macrocyclic chelators have been evaluated for application in nuclear imaging [5,6,7]. The poly (amino) carboxylate ligands, such as DOTA (1,4,7,10-tetraazacyclo-dodecane-1,4,7,10-tetraacetic acid), NOTA (1,4,7-tria azacyclo nonane-1,4,7-triacetic acid), DTPA (diethylenetriaminepentaacetic acid), and poly (aminophosphonic acids), and their derivatives are one of the most frequently used chelators [8,9,10,11,12]. However, these chelators often require harsh conditions for complexation, such as heating of over 80 °C or acidic medium and an excess of ligand. They also exhibit slow reaction kinetics. Thus, the search for new alternative chelating ligands is still a subject in medical inorganic chemistry [7,11,13].

Porphyrins have the ability to chelate metal ions, due to the system four pyrrole nitrogen atoms, which are highly selective for ions with an ionic radius of about 70 pm (i.e., ionic radius of copper (II) is about 72 pm, galium about 67 pm). When the ionic radius of the coordinating cation is in the range 55–80 pm, the metal ion can fit into the center of the planar tetrapyrrolic ring system, forming regular metalloporphyrin. Metal ions with a greater ionic radius (over 80–90 pm) are located out of the porphyrin plane and sitting-atop (SAT) complex is formed [14]. In this case, the reaction is relatively fast and proceeds through a pathway that involves a mononuclear activated complex. Chelating properties are not significantly affected by the type and number of substituents in the ring, allowing tuning of the basic in vivo parameters, such as hydrophilicity/hydrophobicity or partition coefficient octanol/water, and extends the application for the intended purpose of imaging. Porphyrins can thus easily serve as bifunctional ligands coupled with various biomolecules, which allows for specific targeting. 

Complexes, although characterized by high values of stability constants, form with relatively poor kinetics, which has led to limited application in the past [15]. Recent advances have resulted in the development of rapid complexation methods with successful applications even with the relatively short-lived isotopes of galium.

Due to the strong complexing properties and catalytic behavior of metalloporphyrins, they have found numerous applications in analytical chemistry and as models for the synthesis of related macrocyclic systems [16,17,18,19]. The recent advances in porphyrin-based materials for metal detection were summarized by Qi et al. [20]. Porphyrins also offer good potential for biomedical and imaging applications as they are potent fluorophores, biologically compatible, and are known to preferentially accumulate in tumor tissue, which is a highly desirable feature for anticancer therapy [20]. One of the most important applications of porphyrins is photodynamic therapy (PDT) of cancer, where the combination of light and a photosensitizer generates active oxygen species close to the tumor to damage the diseased tissues [21,22]. Porphyrin-based molecules have also been used in in vivo magnetic resonance imaging (MRI) [23,24] and photodynamic antimicrobial chemotherapy [25]. They can also effectively deliver and release some drug molecules to the targeted cell, which improves therapies for a range of human diseases [25,26]. Chelated metals can also provide a convenient handle for bioconjugation with other molecules via axial coordination. Meso-tetrakis(4-sulpho natophenyl)porphyrin modified with gold nanoparticles exhibits high efficiency in binding with doxorubicin, a common drug used in a wide range of cancers, and effectively delivers it within the nucleus of tumor cells [27]. 

The aim of this paper is to present a short overview of the radionuclides complexed by porphyrin ligands and used in SPECT and PET. These chelation reactions are discussed in terms of the complexation conditions and kinetics and the stability of the formed complexes. 

## 2. The Main Radiometals for PET and SPECT

The choice of radionuclide for diagnostic radiopharmaceuticals in PET and SPECT mainly depends on its nuclear properties, such as the type of radiation, half-life, and energy. Other factors may include radionuclide production, the conditions for radiolabeling, specific activity, and coemission of other gamma rays, which unnecessarily increase the radiation dose. ^11^C (t_½_ = 20 min) and especially ^18^F (t_½_ = 109 min) are widely used in PET, but their relatively short half-lives and hard labeling conditions are not suitable for use for labeling with antibodies and peptides [28]. For this reason, ^64^Cu, ^99m^Tc, and ^89^Zr have received increasing attention in the last years [29,30,31,32,33,34,35]. Among them, ^68^Ga and ^99m^Tc (also ^44^Sc) are available from the appropriate generator, which enables wider application. Another important trend is the use of appropriate isotope pairs (with similar biological distribution) for both therapeutic and diagnostic purposes (so-called theranostics) [29,30,32,36]. One radionuclide is a β^+^ or γ emitter that allows the diagnostics while the second emits radiation (α, β, or Auger electrons), which is able to destroy malignant cells. The pairs of ^64^Cu/^67^Cu, ^68^Ga/^177^Lu, ^64^Cu/^177^Lu, and ^44^Sc/^47^Sc could be applied as diagnostic and therapeutic agents, respectively. 

One of the most interesting radionuclides for PET is ^64^Cu. It undergoes multiple decay paths, allowing not only PET imaging through positron emission but also offering the possibility of treatment due to the emission of β^−^ radiation. It also allows for online monitoring of targeted therapy. Another advantage of ^64^Cu is its low positron energy (653.1 keV) with a short average tissue penetration range (0.7 mm), which increases the resolution of the obtained images [30]. Its relatively long half-time (t_½_ = 12.7 h), compared to other popular PET isotopes, offers an opportunity for the labeling of biomolecules, such as proteins, antibodies, and peptides, which require more time to reach target tissues or processes. ^64^Cu is commonly produced in class 2 (up to 20 MeV per particle, proton, or deuteron beam) medical cyclotrons utilizing the ^64^Ni (p,n) ^64^Cu reaction. The well-established coordination chemistry of copper allows for its reaction with a wide variety of chelators that can potentially be linked to antibodies, proteins, peptides, and other biologically relevant molecules.

The availability of ^68^Ga from a generator in which gallium is constantly produced and that it can be eluted from resin containing the long-lived parent ^68^Ge nuclide has led to a rapid increase in the use of this radionuclide. Particularly, after the approval by the Food and Drug Administration of ^68^Ga-DOTATATE and ^68^Ga-DOTATOC, peptides with a covalently bonded DOTA bifunctional chelator have been used to image neuroendocrine tumors [37]. In terms of decay properties, ^68^Ga provides high positron abundance (89%, 1.83 MeV) and a half-life of 68 min, which is compatible with the pharmacokinetic profile of most small molecule imaging agents [32]. The production of ^68^Ga is also possible using the ^68^Zn (p,n) ^68^Ga nuclear reaction with a medical cyclotron [38]. As the formation of ^67^Ga and ^66^Ga isotopes is also possible during this reaction, the radionuclidic specification based on European Pharmacopeia for cyclotron-produced ^68^Ga recommended a maximum content of these isotopes of 2%. 

^67^Ga, a cyclotron-produced radiometal via the ^68^Zn (p,2n) ^67^Ga nuclear reaction with a half-life of 78.3 days, is used in SPECT. The most abundant emitted γ-photons relevant to imaging have energies and relative abundances of 93 (relative abundance of 39%), 184 (21%), and 300 keV (17%) [2]. Due to its longer half-time in comparison to ^68^Ga, ^67^Ga does not have kinetic constraints under radiolabeling conditions. Thus, mild radiolabeling conditions suitable for sensitive biomolecules can be used. As the biological iron transporter transferrin has a strong affinity for Ga (III) [39], its complexes must be sufficiently inert to transchelation. The similar biochemical properties of ^67,68^Ga diagnostics can be combined with therapy using ^177^Lu [40]. 

^89^Zr, with its relatively long half-time (t_½_ = 78.4 h) and low positron energy (396 keV), has received attention for radiopharmaceutical development due to its favorable nuclear decay properties that make it useful in the labeling of antibodies for immuno-PET applications [34]. It is mainly produced in medical cyclotrons via the proton irradiation of natural yttrium foils. Separation from other metal impurities is usually carried out using anion exchange chromatography due to the high affinity of Zr(IV) for hydroxamate-based resins under acidic conditions [41]. Lin et al. proposed a semi-automated approach for the production of ^89^Zr-oxalate/^89^Zr-chloride with high effective specific activity [42].

^44^Sc, a positron-emitting isotope, is of particular interest in PET imaging, and more broadly for theranostic applications in conjunction with ^47^Sc, which emits β^−^ [43]. ^44^Sc can be produced by proton bombardment of ^44^Ca in cyclotrons. It can also be obtained from a ^44^Ti/^44^Sc generator, but difficulties in the production of parent ^44^Ti makes it less affordable than the ^68^Ge/^68^Ga generator. ^44^Sc has biochemical properties that are similar to ^68^Ga, but its half-life is almost 4 times longer (t_½_ = 4.04 h), with an average positron energy of 632.0 keV; thus, this makes it suitable for the imaging of longer biological processes, such as protein metabolism [44].

^99m^Tc is the most common nuclide for SPECT due to its physical and chemical characteristics [45,46]. It is a metastable isomer of ^99^Tc, to which it de-excitates emitted gamma photons. ^99m^Tc is available from the ^99^Mo/^99m^Tc generator or is produced in a cyclotron via proton bombardment of ^100^Mo. Its half-life is equal to 6.01 h. This allows for easy preparation and administration of the radiopharmaceuticals and a gamma ray energy of 140 keV is suitable for detection [33]. The possibility of the occurrence of technetium in several oxidation states allows the incorporation of this radionuclide into a variety of functional groups, which can be specifically adapted to different organs, but its redox state has to be carefully maintained to prevent release from the complex. Moreover, these properties allow the formation of its metallic fragments, also named cores or moieties [47,48,49]. The most known examples of these cores are Tc-oxo [TcO(H_2_O)_4_]^3+^, Tc-dioxo [TcO_2_(H_2_O)_4_]^+^, Tc-nitrido [TcN(H_2_O)_4_]^2+^, Tc-HYNIC (where HYNIC = 6-hydrazino nicotinamide), and Tc-tricarbonyl [Tc(CO)_3_(H_2_O)_3_]^+^ presented in Figure 1. They are prepared after the reduction of TcO_4_^−^ to a suitable oxidation state, predominantly using SnCl_2_ in acidic media.

Another radioisotope that haas received attention is ^111^In. It is produced in a cyclotron using the ^112^Cd (p, 2n) ^111^In reaction. The energies of the γ-ray emissions (171 and 245 keV) are higher than that of ^99m^Te and are widely employed in SPECT for tumor imaging. The half-life of ^111^In (2.8 days) is especially suited to the imaging of antibodies that tend to have longer biological half-lives, such as lymphocytes, platelets, monoclonal antibodies, and many others. Bétak et al. proposed another production method of ^111^In from its grandparent ^111^Sb [50]. The latter is formed after proton bombardment of an enriched ^112^Sn target via the ^112^Sn (p, 2n)^111^Sb reaction. Moreover, this reaction leads to the parent nuclide ^111^Sn, which decays to ^111^In.

^51^Mn, ^52g^Mn, and ^52m^Mn are radioactive isotopes of manganese that emit positrons and are used in PET imaging [51,52,53]. ^51^Mn is produced in a cyclotron via the ^68^Zn(p,2n)^67^Ga nuclear reaction. Its half-life of 46 min decays by 97% by β^+^ emission, but the positron energy is rather high at 2.2 MeV. ^52g^Mn is also produced by the ^nat^Cr(p, n)^52g^Mn nuclear reaction in lower energy ranges from 16.9 to 8.2 MeV. The isolation of radiometal from the chromium target is well developed and is successful using ion chromatography methods, obtaining the product in the form of [^52^Mn]MnCl_2_. The relatively long half-life of 5.59 days of ^52g^Mn is advantageous for purification, synthesis, and the investigation of longer-term biological processes. However, only 29% of positron decay occurs and three high-energy photons are emitted during decay, which limits its clinical relevance. ^52m^Mn (t_½_ 21 min, β+ 96.6%) also has high-energy gamma emission. Its short half-life is hardy compatible with the available time-consuming target separation methods. In addition, it emits a photon with relatively high energy (1.022 MeV), which causes pair formation and thus incorrect PET images.

In addition to the above described radiometals, there are several other positron emitters, such as ^86^Y (t_½_ = 14.7 h), E_β+_ = 1250 keV [54,55], ^62^Zn (t_½_ = 9.3 h) [56], and ^57^Co with a half-life of 17.5 h and E_β+_ = 570 keV [57], which have been proposed for in vivo PET imaging. 

Table 1 summarizes the nuclear properties of the main radiometals used for PET and SPEC imaging.

## 3. Porphyrins as Ligands for Radiometals

### 3.1. Copper

Several research groups have explored the use of porphyrin ligands with different functional groups as potential chelators for ^64^Cu-based radiopharmaceuticals due to their ability to form stable complexes under physiological conditions. The advantage of this approach is the interaction of porphyrins with tumor cells, the minimal toxicity of the ^64^Cu–porphyrin complex, and that the chelation reaction does not alter the biodistribution and pharmacokinetics of the host porphyrin molecules [58,59,60,61,62,63]. The structures of the studied porphyrin ligands are presented in Figure 2.

Mukai et al. [58] studied the chelation reactions of various porphyrins, such as protoporphyrin IX (PPIX), 5,10,15,20-tetrakis(4-aminophenyl)porphyrin (TAPP), 5,10,15,20-tetrakis(4-sulfophenyl)porphyrin (TSPP), and 5,10,15,20-tetrakis(4-carboxy phenyl)porphyrin (TCPP), with ^64^Cu in acetate buffer (pH 6.0) containing 0.5% (*w*/*v*) Tween 20. Intensive heating was required for each porphyrin: 60 min at 50 °C for PPIX and TSPP; 5 min at 100 °C for TAPP; and 60 min at 100 °C for TCPP. The addition of ethanol increased the efficiency of chelation from 58.6% to 80.4% for TCPP and from 19.7% to 68.6% in the case of TSPP. The authors also synthesized [^64^Cu]PPIX-PEG6-BBN conjugate, which contains bombesin (BBN) analog, a peptide that interacts with the gastrin-releasing peptide receptor (GRPR) attached to the PPIX molecule through PEG6 spacer. It showed significantly higher uptake of PC-3 (prostate cancer) cells than ^64^Cu-labeled PPIX. Moreover, it was found that ethanol is a good radiolytic stabilizer for labeling [58].

Faster formation of Cu–TCPP complex was obtained in the presence of some reducing agents such as hydroxylamine, ascorbic acid, or morin (a biologically active natural antioxidant that occurs widely in plants) [59]. As the ionic radius of Cu(I) is significantly larger than that of Cu(II), the formation of SAT complex involves more favorable kinetics. The best reaction rate was achieved in borate buffer at pH 9 and a ratio of the reactants Cu:TCPP:ascorbic acid of 1:1:10. Under these conditions, the reaction was almost immediate, below 1 min (Figure 3). The chosen reducing agent is safe and can be used in radiopharmaceutical applications.

The reaction of ^64^Cu with 5-(4-aminophenyl)-10,15,20-triphenyl)porphyrin was conducted in DMSO at 37 °C for 60 min with the use of acetate buffer at pH 5.65 [60]. Ascorbic acid was used for the reduction of Cu(II) to Cu(I) to take advantage of the SAT complex, with a labeling yield > 95%. The complex showed high resistance towards transchelation and 85% of the complex remained intact in an excess of EDTA after 48 h of incubation. It also exhibited good stability (>95%) for up 48 h when incubated with human serum. Animal PET studies showed rapid clearance of the complex from healthy mice and rats, which is beneficial in the view of further research.

Fazaeli et al. [61] synthesized ^64^Cu complex with 5,10,15,20-tetrakis(pentafluoro phenyl)porphyrin (TFPP) in the presence of acetate buffer at pH 5.5. The addition of fluoride groups in the periphery of the porphyrin increased the hydrophilic character of the ligand and its solubility in water. The mixture of the reagents was refluxed at 100 °C for 60 min. Incubation of the ^64^Cu–TFPP complex in human serum showed no loss of radionuclides for up to 2 days. Imaging showed quick (4 h) renal clearance while the injection of free ^64^CuCl_2_ resulted in higher liver uptake.

Recently, porphyrin-based nanomaterials have received much attention due to their excellent imaging capacities [64,65,66,67]. ^64^Cu-labeled pentafluorophenylporphyrin complex was successfully grafted onto mesoporous silica functionalized with 3-amino propyltrietoxysilane groups [66]. Its biodistribution in fibrosarcoma-bearing rats showed high tumor uptake and fast excretion from the body. Other authors prepared polyethylene glycol (PEG)-modified TCPP nanoparticles labeled with ^64^Cu that could help to evaluate renal clearance [67,68]. The scheme for their synthesis with various molecular weights of PEG chains is presented in Figure 4. ^64^Cu–TCPP–PEG nanoparticles with a larger molecular weight (30K) showed higher tumor uptake due to an enhanced permeability and retention effect, while the lower ones (2K) were more suitable for renal clearance. The prepared radiotracers were found to be highly stable in serum for 48 h, even in the presence of NOTA competitive conditions; thus, they are suitable for in vivo PET imaging. 

Luo et al. proposed the synthesis of a multifunctional system based on poly (vinyl alcohol)–porphyrin conjugate, which was labeled with ^64^Cu [69]. The PVA–porphyrin conjugate was prepared through ester formation, the polymer was then dissolved in DMSO, and dialysis was performed against water. A chelation reaction with ^64^Cu was carried out by simple stirring of the reagents at room temperature for 2 h. PET imaging showed that ^64^Cu-labeled poly (vinyl alcohol)–porphyrin nanoparticles started to accumulate at tumor sites 16 h after injection. This theranostic nanoplatform integrates cancer optical imaging, positron emission tomography, photodynamic and photothermal therapy, and drug delivery functions in one formulation [63,70].

Fan et al. [71] proposed a porphyrin-based molecule for multimodal tumor imaging. The ^64^Cu–Pyro–3PRGD2 molecule combining a porphyrin derivative, an RGD dimer peptide (3PRGD2), and ^64^Cu, exhibited high tumor specificity in both positron emission tomography and optical imaging in vivo. Additionally, a highly hydrophilic polyethylene glycol (PEG) chain as the linker between the porphyrine macrocycle and peptide ligand increased the water solubility of the conjugate. The tripeptide Arg-Gly-Asp (RGD) is a high-affinity ligand of integrin α_v_β_3_ targeting the RGD-conjugated molecular probes or nanoparticles to α_v_β_3_-overexpressing cancer cell lines. 

### 3.2. Gallium

Ga(III) and Fe(III) have the same ionic charge, similar ionic radii (62 pm for Ga(III) and 65 pm for Fe(III), respectively), and both tend to form six-coordinated complexes [71,72]. Due to this similarity, ligand exchange with the abundant blood serum protein transferrin can occur in vivo, resulting in lung, liver, and bone accumulation of ^68^Ga. Precipitation of Ga(OH)_3_, which starts at pH > 3, makes radiolabeling difficult. The presence of weakly coordinating anions (citrate, acetate, or oxalate) can prevent this undesirable process [72]. The presence of metal ion impurities, such as Cu(II), Fe(III), or Pb(II), in the generator eluent could reduce the yield of ^68^Ga labeling [73]. Complexation of gallium ions within the porphyrin core often requires reaction temperatures above 100 °C, which is unsuitable for temperature-sensitive moieties. Gallium–porphyrin complexes are suitable for the development of agents for theranostic applications involving tumor diagnosis using PET and PDT for targeted tumor therapy [74,75].

The complexation of ^68^Ga with tetrapyrrole derivatives, such as hematoporphyrin (HP), protoporhyrin IX (PPIX), and tetraphenylporphyrin (TPP), was conducted by Zoller et al. [76,77]. PPIX is a native porphyrin derivative present in the body as part of the heme protein. As labeling experiments of water-soluble HP and PPIX porphyrins in a water/acetone mixture and heating in an oil bath at 90 °C for 15 min produced a low reaction efficiency, microwave-enhanced radiosynthesis was applied. Using this approach, labeling yields of 69% for ^68^Ga-HP after 5 min and 49% for ^68^Ga-PPIX after 7 min were obtained (Figure 5). Complexation with liphophilic TPP was achieved in chloroform solution using anhydrous conditions via the indirect nuclear reaction with ^68^Ga-acetyloacetone as the labeling agent [69]. This resulted in a labeling yield of 82% after 5 min with microwave irradiation. Transchelation of ^68^Ga to DTPA solution or *apo*-transferrin was not observed over a period of 2 h.

As was mentioned above, the introduction of substitutes on the peripheral positions of the porphyrin core increases the hydrophilicity of porphyrin derivatives to promote renal clearance over hepatobiliary clearance [78]. Some hydrophilic porphyrin derivatives have been proposed for complexation with ^68^Ga, such as 5,10,15,20-tetrakis(pentafluorophenyl)porphyrin (log P_o/w_ = 0.62) [79], 5,10,15,20-tetrakis(2,4,6-tri methoxyphenyl)porphyrin (log P_o/w_ = −1.14) [80], 5,10,15,20-tetrakis(*p*-carboxy- methyleneoxyphenyl)porphyrin (log P_o/w_ = −0.25) [75], and 5,10,15,20-tetrakis(4-methyl-pyridyl)porphyrin (log P_o/w_ = −4.3) [81]. Complexation reactions with Ga(III) were conducted in the presence of acetate buffer in a boiling water bath for a period in the range of 15 [74] to 60 min [79]. 

Pan et al. synthesized the water-soluble bimetallic gallium–porphyrin–ruthenium–bipyridine complex (^68^GaporRu) with an 85% yield using microwave irradiation for 15 min [82]. Its structure is presented in Figure 6. In comparison with a similar complex with zinc (ZnporRu), gallium complex also inhibits cancer cells’ growth in their early stages. The acidity of ^68^GaporRu (pK_a_ = 3.45) enables specific subcellular localization in the lysosome, while ZnporRu exhibits mitochondria specificity. Thus, it was considered a novel functional bioprobe for PET imaging and a photodynamic therapy agent.

The effects of structural variation and the number of positive charges in the tetracationic and tricationic porphyrin derivatives on the tumor targeting efficacy were studied by Guleria et al. [83]. In vivo experiments in a tumor-bearing animal model revealed a relatively longer retention of tetracationic ^68^Ga-labeled porphyrin in the tumor lesion compared to the of ^68^Ga-labeled tricationic derivatives.

An extended study of neutral, polycationic, and polyanionic metalloporphyrins labeled with ^68^Ga showed extremal flexibility for modifications of the porphyrin cores [84]. The addition of nitroimidazole or sulfonamide groups, which were used as vectors, improved the pharmacokinetics of porphyrin tags and the stability in serum. However, it did not influence the fluorescent properties, allowing in vitro confocal studies and visualization. The conjugation of porphyrins with peptides was also achieved and provided effective targeting of the overexpressed receptors on tumor cells. Porphyrin probes were successfully tested as the bifunctional chelator scaffolds for PET with ^68^Ga and for SPECT as the central metal ions.

### 3.3. Technetium

Due to the difficulties in the chelation of ^99m^TcO_4_^−^, obtained from a molybdenum generator after elution by saline solution, with a porphyrin core, Wang et al. proposed the use of acetylacetone (acac) as a conjugator to first form ^99m^Tc(acac) complex (refluxing at 150 °C for 30 min with slow nitrogen flow) and then labeling with 5,10,15,20-tetrakis(4-carboxyphenyl)porphyrin (TCPP) [33]. The labeling efficiency of the formed complex was about 99% and log *P* was equal to -0.86, showing its hydrophilic nature. As most of the radioactivity accumulated in the liver, the ^99m^Tc(acac)–TCPP complex seems to suitable as an imaging agent for this organ.

### 3.4. Zirconium

Hexadentate siderophore desferrioxamine (DFO) is mainly used for ^89^Zr chelation in immuno-PET imaging [33,34]. However, zirconium complexes with DFO are partially unstable and released radiometal can accumulate in bone tissue. It was explained that DFO occupies only six coordination sites while Zr(IV) forms octacoordinated complexes. For this reason, alternative bifunctional chelating agents are being evaluated, mostly containing hydroxamate coordinating units [85,86,87].

The reaction of Zr-acetylacetone with *p*-methoxy-meso-tetraphenylporphyrin [88] and meso-tetraphenylporphyrin [89] in the presence of phenol and its derivatives in chloroform (conducted at 200–220 °C over a salt bath) resulted in the formation of corresponding axially ligated complexes [Zr(*p*-OCH_3_TPP)(Y)(X)] and [Zr(TPP)(Y)(X)], where Y = acac and X = phenolates. The coordination number of zirconium in both complexes was reported as seven, and due to the large ionic radius of Zr(IV), the metal is out of the plane of the porphyrin ring. There is no research on the application of these complexes in nuclear medicine imaging techniques. It was reported that among all the complexes studied, Zr(TPP)(acac)(*p*-NO_2_PhO) showed the highest antibacterial sensitivity against the bacterial strains [88]. 

### 3.5. Other Radiometals

Complexation of ^111^In with 5,10,15,20-tetrakis(3,5-dihydroxyphenyl)porphyrin, TDHPP), 5,10,15,20-tetrakis (4-hydroxyphenyl) porphyrin, THPP), and 5,10,15,20- tetrakis (3,4-dimethoxyphenyl) porphyrin, TDMPP) was prepared for SPECT imaging [90]. The mixture of reagents was heated at 80 °C for 60 min in acetic buffer. They showed more than 99% radiochemical purity and no loss of radionuclides in freshly prepared human serum over 2 days at 37 °C. The partition coefficients (calculated as log P_o/w_) for ^111^In-TDHPP, ^111^In-THPP, and ^111^In-TDMPP were 0.88, 0.8, and 1.63, respectively; thus, the dihydroxy complex of ^111^In-TDHPP showed more hydrophilicity compared to the mono-hydroxyl compounds. The obtained complexes accumulated mainly in the liver and kidney of the rat tissues, which are typical accumulation sites of porphyrins. 

Under similar reaction conditions (heating at 100 °C for 60 min), ^111^In was labeled with 5,10,15,20-tetrakis(pentafluorophenyl)porphyrin, TFPP) [91]. The octanol/water partition coefficient for this complex was found to depend on the pH of the solution and at pH 7, the log P_o/w_ was 0.69. For better comparison, a biodistribution study was also performed on free ^111^InCl_3_ solution in wild-type rats. The indium cation was rapidly removed from the circulation, and accumulated in the liver, and a major fraction was excreted slowly in 24 h through the kidney, in an almost steady manner. The ^111^In–TFPP complex also accumulated in the kidney but additionally in the spleen, gradually up to 15%, and its excretion significantly increased after 24 h.

Tamura et al. developed multicomponent PET tracers based on PDT agents by labeling ^62^Zn with glycosylated 5,10,15,20-tetrakis(pentafluorophenyl)porphyrin, which has S-glycosylated groups (Figure 7) [56,92]. The total time required from the synthesis (heating at 60 °C for about 10 min, concentrated under reduced pressure, and dissolved in EtOH/PEG-400/water mixture in a volume ratio of 2:3:5) to administering them into mice was less than 30 min. The cellular uptake and cancer cell-selective accumulation of these complexes depend on the numbers of S-glycosylated groups and their orientation (cis or trans), being the highest for trans isomer with two groups. Their distribution in the blood was maintained over 24 h and slightly decreased in the liver, kidney, and spleen. 

Manganese porphyrins are used as paramagnetic contrast agents, with low toxicity, high electronic spins, a fast water exchange rate, and high complex stability [93,94,95]. The most extensively investigated, Mn(III) tetraphenyl porphyrin sulfonate (MnTPPS_4_), exhibited no demetallation in vitro in human plasma for up to 9 days and only about 1% degree of demetallation in vivo in the liver and kidney up to 4 days post administration [95]. Klein et al. demonstrated the possibility of labeling MnTPPS_4_ with the no-carrier-added positron emitter ^51^Mn [96]. Thus, the labeled compound could allow non-invasive determination of the pharmacokinetics in humans and additionally could serve as a new tumor-localizing radiopharmaceutical. The complex formation kinetics were investigated, and the apparent rate constants were determined as 0.0244 s^−1^M^−^^1^ at 44 °C and 5.9 s^−1^M^−^^1^ at 108 °C. Gawne et al. described a new method for the radiochemical synthesis of ^52^Mn–porphyrin complexes using six porphyrin ligands with various lipophilicities and charges and then assessed their liposome labeling properties [97]. Using a microwave synthesizer and heating at 165 °C for 1 h, radiochemical yields > 95% were achieved at a ligand concentration of 0.6–0.7 mM. In contrast, heating at 70 °C for 1 h without microwave irradiation resulted in low radiochemical yields (0–25%) and most porphyrins did not reach completion after 24 h.

## 4. Conclusions

Nuclear medical imaging is a field of molecular diagnostics that is still developing. There is a continuous need for new efficient chelators that can satisfy all the requirements. Porphyrins are promising candidates for ligands in radiopharmaceutical development. They form stable complexes in vivo, which prevents the release of radionuclides. As porphyrins have an affinity for tumor cells, they can accumulate in tumor tissue, allowing for accurate imaging. The properties of the designed radiotracer can be easily adjusted through modification of the peripheral functional groups, as it has little effect on the labeling efficiency. This article provides an overview with great perspective for further development of the next generations of PET radiopharmaceuticals.

## Figures and Tables

**Figure 1 molecules-27-03311-f001:**
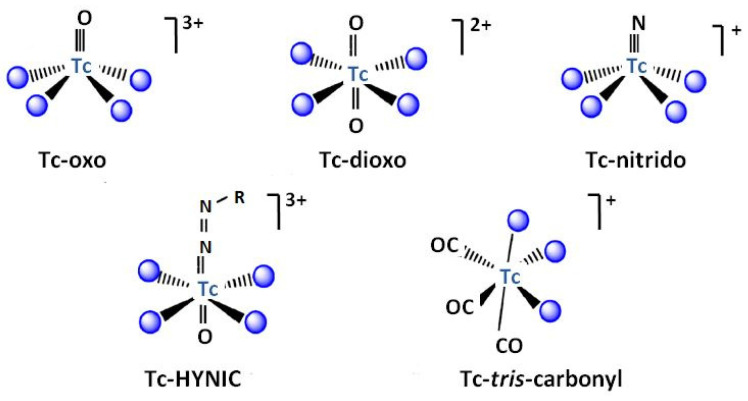
Technetium inorganic fragments used for labeling bioactive molecules. HYNIC- 6-hydrazinonicotinamide. Adopted from [48].

**Figure 2 molecules-27-03311-f002:**
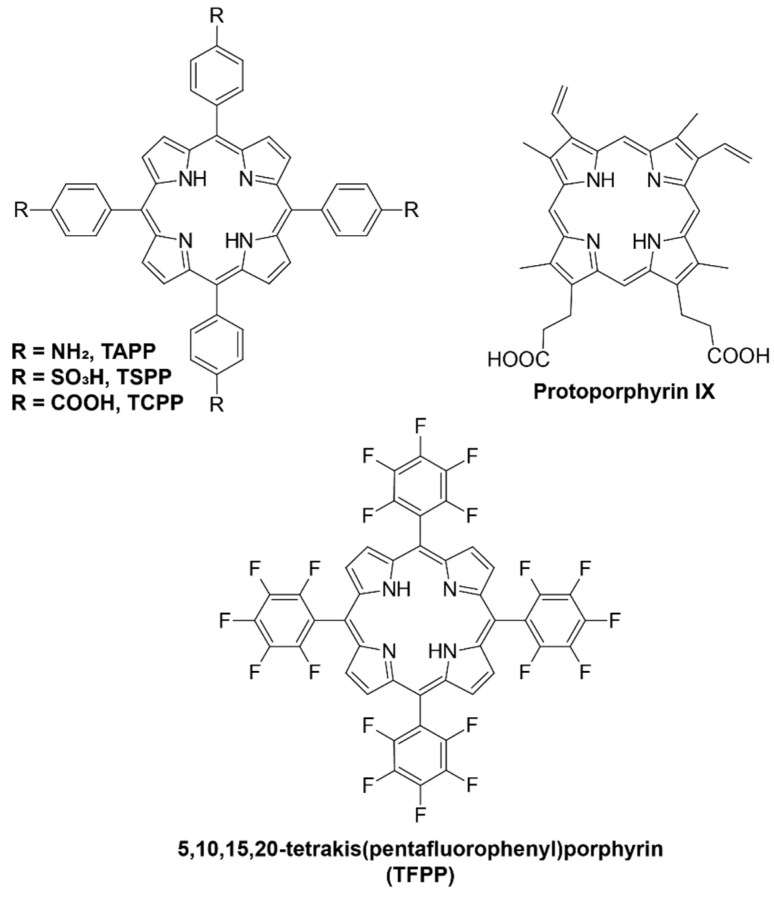
The structures of studied porphyrin ligands complexed with ^64^Cu for use in PET.

**Figure 3 molecules-27-03311-f003:**
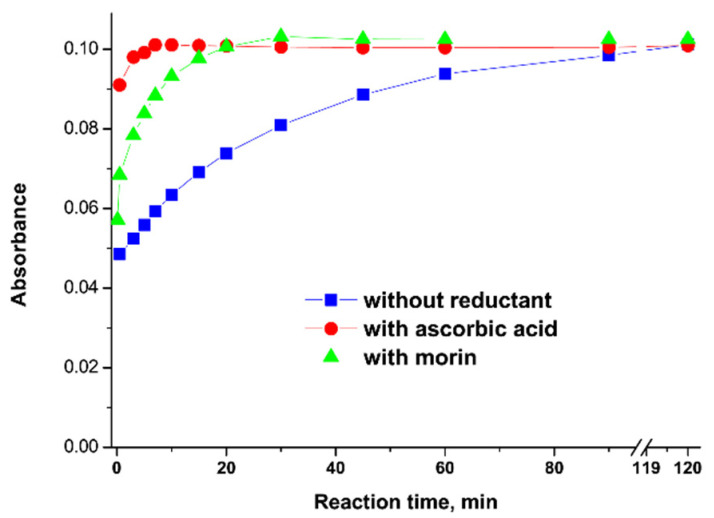
The changes in absorbance at 542 nm (λ_max_ for complex) over time for the reaction of Cu(II) and TCPP in the presence of ascorbic acid and morin. [Cu^2+^] = [TCCP] = [ligands] = 10^−5^ M. Reprinted with permission from [59]. 2016. Copyright Elsevier.

**Figure 4 molecules-27-03311-f004:**
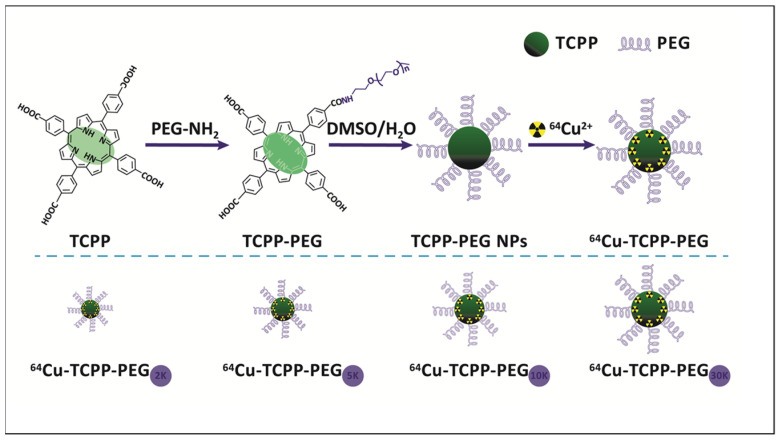
Synthesis of TCPP–porphyrin nanoparticles with various molecular weights of PEG chains. Reprinted with permission from [67]. 2016. Copyright Elsevier.

**Figure 5 molecules-27-03311-f005:**
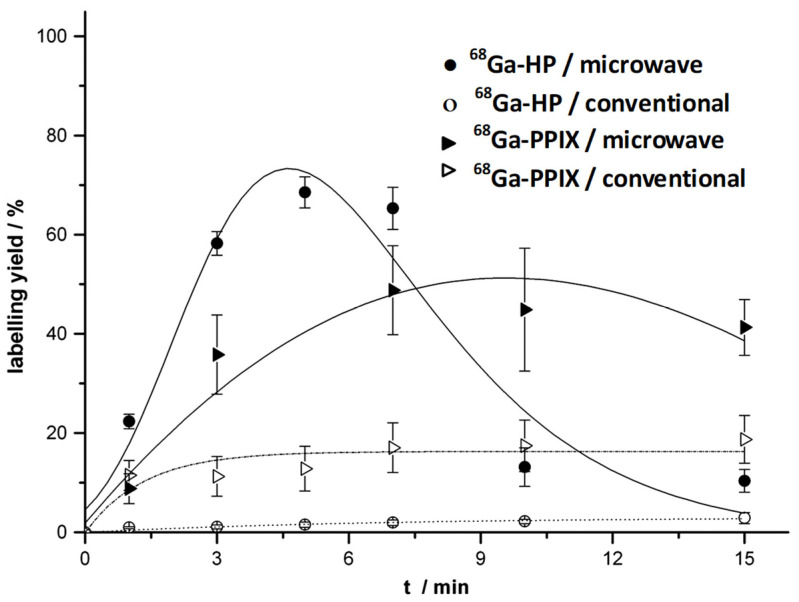
Time dependency of the labeling yield of ^68^Ga-hematoporphyrin (HP) and ^68^Ga-PPIX (protoporphyrin IX) in chloroform solution under conventional conditions (oil bath at 90 °C) and microwave irradiation (170 °C, max. 150 W). Reprinted with permission from [76]. 2013. Copyright Elsevier.

**Figure 6 molecules-27-03311-f006:**
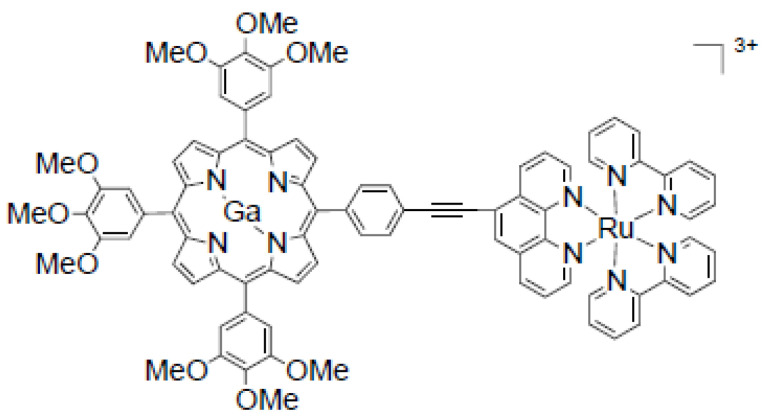
Structure of gallium–porphyrin–ruthenium–bipirydyne complex (GaporRu). Reprinted with permission from [82]. 2016. Copyright American Chemical Society.

**Figure 7 molecules-27-03311-f007:**
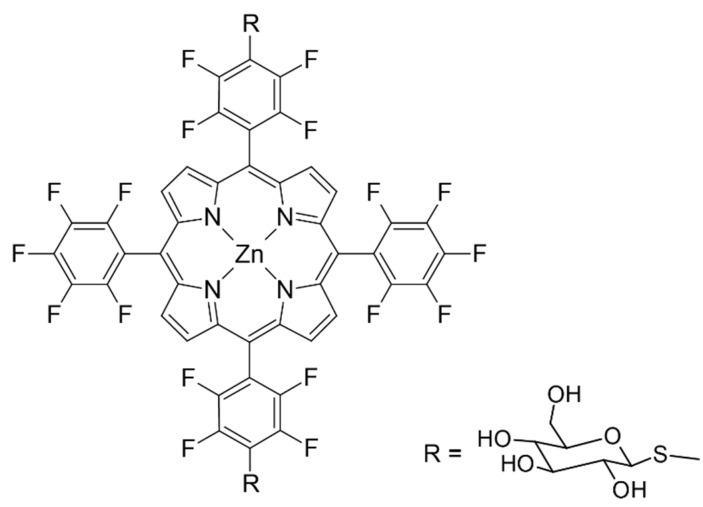
^62^Zn fluorinated porphyrin conjugated to S-glycosylated groups. Adopted from [56].

**Table 1 molecules-27-03311-t001:** The nuclear properties of the main radiometals used for PET and SPEC imaging.

Radionuclide	Source	Reaction	Radiation	t_½_
^44^Sc	cyclotron	^44^Ca(p, n)^44^Sc	β^+^	3.97 h
^62^Zn	cyclotron	^nat^Cu(p, x)^62^Zn	β^+^	9.19 h
^64^Cu	cyclotron	^64^Ni(p, n)^64^Cu	β^+^	12.7 h
^67^Ga	cyclotron	^68^Zn(p, 2n)^67^Ga	γ	3.26 d
^68^Ga	generator	^68^Ge/^68^Ga	β^+^	68 min
	cyclotron	^68^Zn(p, n)^68^Ga		
^89^Zr	cyclotron	^89^Y(p, n)^89^Zr	β^+^	3.3 d
^99m^Tc	generator	^99^Mo/^99m^Tc,	γ	6.0 h
^111^In	cyclotron	Cd(p, xn)^111^In	γ	2.83 d

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
