# Peer review of "Porphyrins as Chelating Agents for Molecular Imaging in Nuclear Medicine"

_molecules, 2022, doi:10.3390/molecules27103311_

Round 1

Reviewer 1 Report

In this review, Pyrzynka et al. describe the use of porphyrins as chelating agents for use in PET and SPECT imaging applications. The review is well-written and could be accepted with the following revisions.

  • The introduction should be revised to give a larger focus on porphyrins and how they are used for PET/SPECT,
  • The review would benefit from a figure or table summarizing porphyrins and the radioisotope they have been radiolabeled with.
  • The review would benefit from examples of in vivo PET/SPECT images from radiolabeled porphyrins.
  • The possibility of multimodal imaging using porphyrins should be mentioned, for example, the theranostic nanocomplexes (Cu-64/Lu-177) for multimodal (PET/IVIS) imaging and therapy prepared by Yu et al (doi: 1002/anie.201710232)
  • Mn-51/52 are other interesting PET isotopes with potential applications using porphyrins that should be mentioned.

Author Response

Title: Porphyrins as chelating agents for molecular imaging in nuclear
medicine

Thank you for the valuable comments regarding the manuscript.

They were taking into consideration and the manuscript was re-written. The added and corrected sentences as well as the new references are marked in red.

Detailed  answers followed.

Introduction part was completed.

Table 1 was added for the nuclear properties of the main radiometals used for PET and SPEC imaging.

In the fragment described the multimodal imaging and therapy using porphyrins, the pair 64Cu/177Lu, except others, was mentioned.

The description of manganese isotopes (e.g. 51gMn, 51mMn, 52Mn) with their potential applications in nuclear medicine using porphyrins was added.

Reviewer 2 Report

I found this review well written and useful material for this field of medical radiations.

I only have few comments and corrections as indicated in the attached copy of the paper

Author Response

Title: Porphyrins as chelating agents for molecular imaging in nuclear
medicine

Thank you for the kindly comments regarding the manuscript.

The added and corrected sentences as well as the new references are marked in red.

The corrections of the indicated phrases was made.

The appropriate reference was added.